# Surgical Management of Scapulothoracic Bursitis in a Patient with Systemic Lupus Erythematosus: A Case Report

**DOI:** 10.3390/jcm12020561

**Published:** 2023-01-10

**Authors:** Jun Ho Choi, Seung Yeon Choi, Kwang Seog Kim, Jae Ha Hwang, Sam Yong Lee

**Affiliations:** Department of Plastic and Reconstructive Surgery, Chonnam National University Hospital, Chonnam National University Medical School, Gwangju 61469, Republic of Korea

**Keywords:** bursitis, bursectomy, scapula, systemic lupus erythematosus, thorax

## Abstract

Scapulothoracic bursitis involves inflammation of the scapulothoracic bursa caused by overuse of the shoulder or traumatic injury. Conservative management is recommended initially, and a surgical approach, such as bursectomy or scapular angle resection, is indicated later. Scapulothoracic bursitis in a patient with systemic lupus erythematosus (SLE) has been rarely reported in the literature. A 34-year-old woman was hospitalized in our hospital for a palpable mass on the right side of her back. She had a history of SLE, which was diagnosed and treated with medication 13 years prior. Chest magnetic resonance imaging (MRI) revealed fluid collection measuring 6.0 cm × 6.0 cm × 2.0 cm between the rib cage and subscapularis muscle. Histopathological examination identified the mass as bursitis with cystic degeneration. Surgical excision was performed at the infrascapular area. About 11 months later, the mass recurred in the same area. Surgical excision was again performed in the same way as before, and the same diagnosis was confirmed. Every 6 months, ultrasound examination is being conducted to assess for recurrence. The patient has not had any further complications or a relapse for the last 3 years. Prompt bursectomy can be a definitive and long-lasting treatment option for scapulothoracic bursitis accompanied by SLE.

## 1. Introduction

Scapulothoracic bursitis is inflammation of the bursa between the scapula and thorax, usually accompanied by pain in the area. The scapula, or shoulder blade, lies above the thoracic wall in the upper back and is associated with many muscles to provide various types of motions at the shoulder joint [1]. The scapulothoracic bursa that affects scapulothoracic articulation is located at the inferior angle of the scapula, between the serratus anterior and chest wall [2]. Inflammation in this structure can result from an acute traumatic event or from situations of chronic overuse, especially in people who are anatomically susceptible to bursa irritation [3]. This is also known as snapping shoulder syndrome, and it usually develops in athletes [3]. However, scapulothoracic bursitis in a patient with a rheumatologic disorder has rarely been reported. A literature search of the Ovid, PubMed, Scopus, and Web of Science electronic databases on 2 August 2022, using the terms “scapulothoracic bursitis”, “systemic lupus erythematosus”, and “rheum”, and not using any date or language restriction, revealed only one previous report [4]. Non-operative management is recommended as the first line of treatment, but this can be ineffective for patients with such a complication. Here, we present a case of a young woman who was previously diagnosed with systemic lupus erythematosus (SLE) and later developed scapulothoracic bursitis, which was successfully treated by surgical excision [5].

## 2. Case Presentation

A 34-year-old woman presented to our clinic complaining of a palpable lump on the right side of her back, along with pain associated with shoulder movements (Figure 1). She had a history of SLE, which was diagnosed 13 years prior, and was on oral prednisolone medication since. Her occupation did not involve extensive use of the shoulders or physical activity in general. At the time of admission, she had developed arthritis on all proximal interphalangeal joints and wrist joints on both hands, along with systemic sclerosis. During the clinical course of lupus, she experienced three lupus flare ups. Her SLE was under control when she first visited our outpatient clinic.

Before visiting our clinic, she was on conservative treatment for right shoulder pain. She was prescribed non-steroidal anti-inflammatory drugs (NSAIDs) along with physical therapy for 6 months to strengthen the rotator cuff muscles. However, her symptoms did not regress, and she was referred to our department for alternative treatment. On physical examination, she had pain in the scapular region when flexing her right shoulder and the intensity was 1–2 as measured by an 11-point pain intensity numeric rating scale. However, the right shoulder was not limited in its range of motion. Enhanced chest magnetic resonance imaging (MRI) revealed fluid collection measuring 6.0 cm × 6.0 cm × 2.0 cm, which was suspected to be scapulothoracic bursitis, between the rib cage and subscapularis muscle (Figure 2). Under general anesthesia, an incision was made on the back to reach the scapular angle. We approached the medial border of the scapula and dissected through the rhomboid major muscle, which was repaired after the excision. The mass was located between the serratus anterior muscle and rib cage (Figure 3). Histopathological examination identified the mass as bursitis with cystic degeneration.

Approximately 11 months later, the mass recurred in the same area. Follow-up enhanced chest MRI showed a mass measuring 10.0 cm × 8.0 cm × 2.0 cm, which was thought to be scapulothoracic bursitis (Figure 4). Another surgical excision was performed in the same manner as before, and the subsequent histopathological examination identified the mass as recurrent bursitis (Figure 5). Despite two surgical excisions, she did not undergo rehabilitation treatment as her shoulder movement remained unrestricted. Every 6 months, ultrasound examination is being conducted to assess for recurrence. The patient has not developed any further complications including pain or recurrence in the last 3 years (Figure 6).

## 3. Discussion

Disruption of the shoulder joint can have a negative effect on the upper body, including on the motion of the neck and head [6]. Snapping shoulder syndrome is a term used to describe clicking and popping of the shoulder when it is lifted. This can be caused by either underlying osseous lesions or inflammation of the scapulothoracic bursae that are present between the scapula and thoracic wall, which prevent touching and friction between the bones. Different approaches to snapping shoulder syndrome should be considered according to the underlying cause. Osseous lesions, such as fibrosis and tumors, can be treated relatively easily, and the prognosis is good after surgical excision [7]. However, when snapping shoulder syndrome is caused by scapulothoracic bursitis, its treatment can be quite difficult.

Studies have recommended starting the treatment of scapulothoracic bursitis with a conservative approach, such as physical therapy, immobilization, and NSAID use [3,5]. Surgical intervention is only considered when conservative treatment fails or when the patient complains of prolonged and unbearable pain due to the condition. Surgical excision is also indicated when snapping shoulder is caused by underlying structural changes, such as osteochondroma [7]. In our case, osteochondroma was ruled out by MRI scans. Occasionally, lupus can cause septic bursitis at the olecranon and prepatellar bursa [8]. We could not rule out the possibility of septic bursitis as the patient previously had episodes of lupus flare ups.

SLE causes various systemic manifestations throughout its clinical course. Patients with SLE are advised regularly undergo monitoring and examinations for disease activity [9]. Lupus flares, which are accompanied by worsening of symptoms, can affect the skin, heart, kidney, muscles, joints, and other parts of the body. Since lupus is an autoimmune disorder, flares can trigger inflammatory conditions throughout the body. These mainly occur in the lungs and fingers [10], and have not been previously reported in the scapulothoracic bursae.

Considering the unique anatomical location of the scapulothoracic bursa, it is very difficult to accurately scan it with an ultrasound probe. It is often necessary to perform dynamic ultrasound scanning during active/passive shoulder movements to uncover the thoracic wall under the scapula. Therefore, dynamic ultrasound protocols for the shoulder have been introduced [11]. However, in our case, additional dynamic scans were not performed because MRI is a better diagnostic tool than ultrasound for determining the anatomy of the scapulothoracic region and diagnosing scapulothoracic bursitis.

In our case, conservative treatments, such as physical exercise and NSAID use, were recommended when scapulothoracic bursitis was first diagnosed. The patient did not show any signs of a lupus flare, and laboratory analysis found that levels of antibodies to dsDNA, C3, and C4 were all within the normal ranges. The chief complaints were a palpable mass and recurrent pain around the right shoulder. As regaining full function of the shoulder was not a priority for our patient, surgical correction was chosen over conservative treatment. Although the bursitis recurred 11 months after the first surgery, the patient did not experience another recurrence after the second surgery, and the outcome has been satisfactory ever since.

## 4. Conclusions

Managing scapulothoracic bursitis in a patient with no underlying medical condition can be challenging in itself, and the further presence of SLE substantially complicates achieving a treatment goal by conventional means. Therefore, patients who have a rheumatologic disorder as the underlying cause for bursitis and who do not require preservation of full shoulder motion should be considered for surgical treatment. Our case demonstrates that surgical bursectomy can be a long-lasting and definitive treatment for scapulothoracic bursitis in patients with SLE.

## Figures and Tables

**Figure 1 jcm-12-00561-f001:**
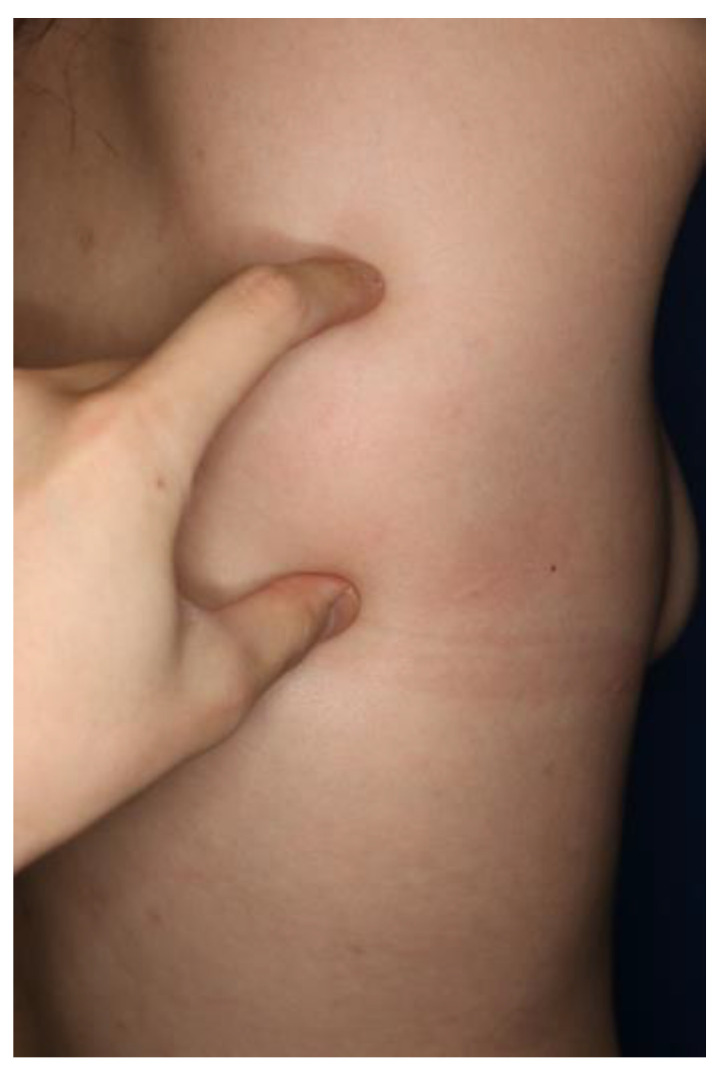
Preoperative photograph of a 34-year-old woman with a palpable mass on the right side of her back.

**Figure 2 jcm-12-00561-f002:**
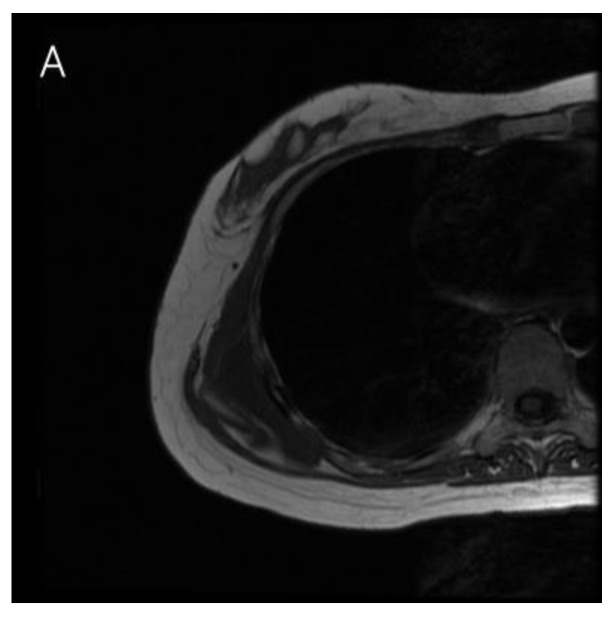
Right shoulder enhanced magnetic resonance imaging showing soft tissue fluid collection (6.0 cm × 6.0 cm × 2.0 cm) in the right posterolateral chest wall, which was suspected to be scapulothoracic bursitis. (**A**) T1-weighted axial spin echo view. (**B**) T2-weighted axial fast spin echo view.

**Figure 3 jcm-12-00561-f003:**
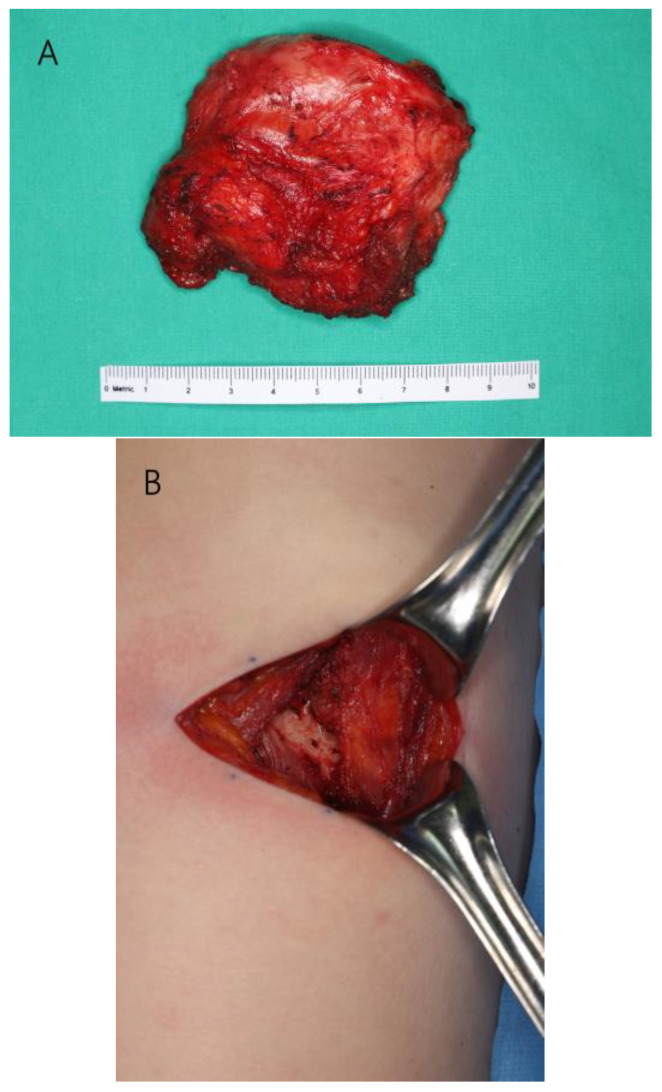
Intraoperative photographs during the first excision. (**A**) Excised mass. (**B**) Scapulothoracic joint space after mass excision.

**Figure 4 jcm-12-00561-f004:**
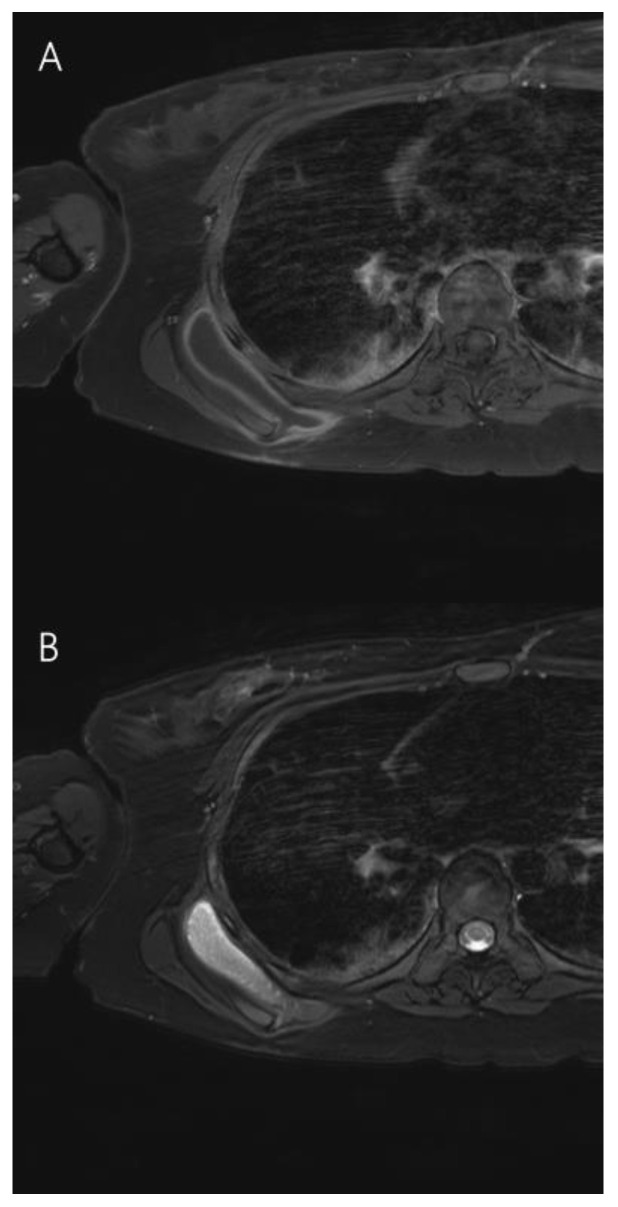
Right shoulder enhanced magnetic resonance imaging showing recurrent bursitis (10.0 cm × 8.0 cm × 2.0 cm) in the right posterolateral chest wall. (**A**) Contrast-enhanced T1-weighted axial view. (**B**) Fat-suppressed contrast-enhanced T2-weighted axial view.

**Figure 5 jcm-12-00561-f005:**
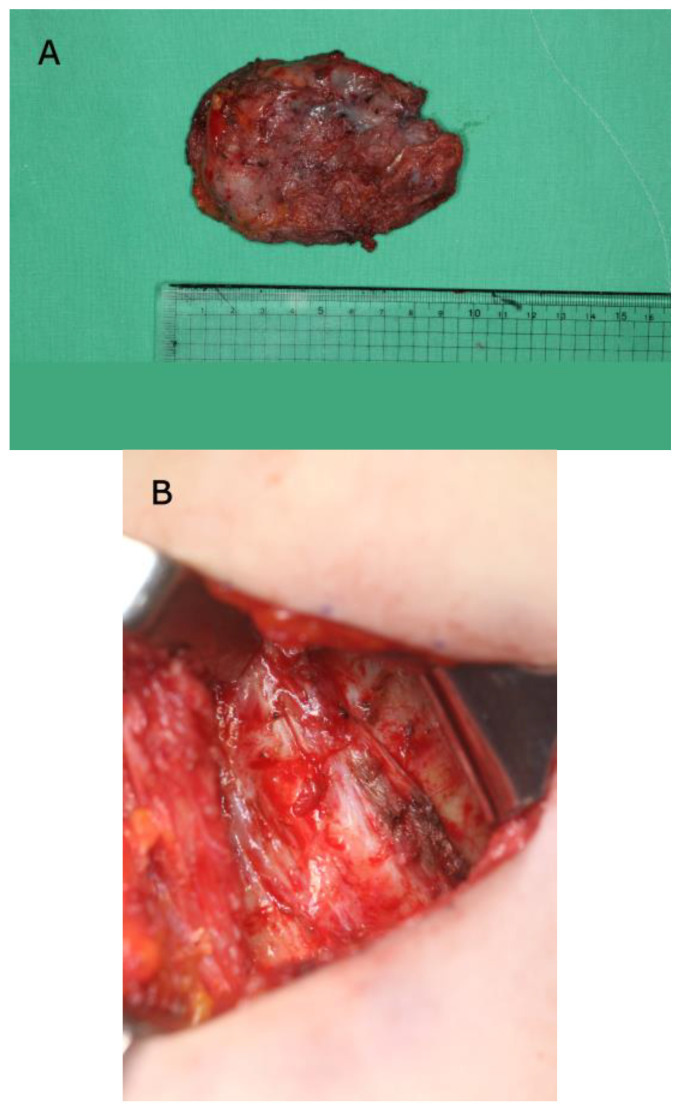
Intraoperative photographs during the second excision. (**A**) Excised mass. (**B**) Scapulothoracic joint space after mass excision.

**Figure 6 jcm-12-00561-f006:**
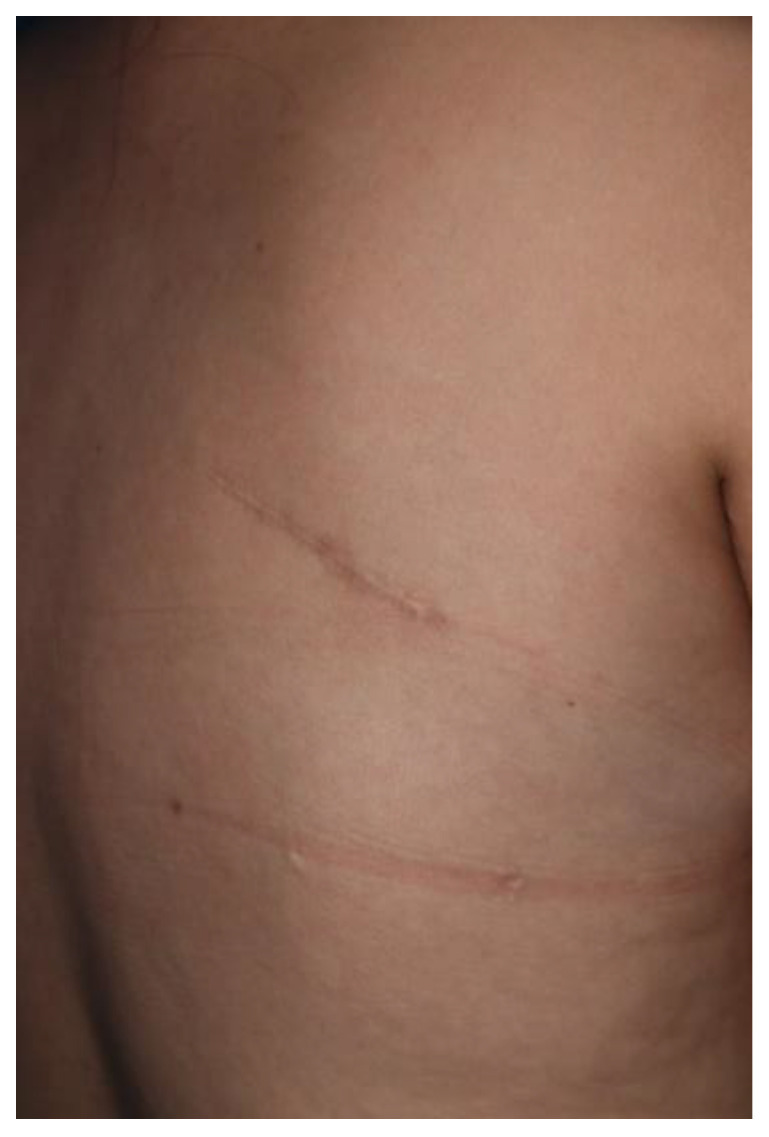
Image obtained 3 years after the second excision. There is no recurrence.

## Data Availability

All data created or analyzed in this study are included in this published article.

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
