# Peer review of "Surgical Management of Scapulothoracic Bursitis in a Patient with Systemic Lupus Erythematosus: A Case Report"

_jcm, 2023, doi:10.3390/jcm12020561_

Round 1

Reviewer 1 Report

The manuscript should be must be improved. There is a lack of relevant clinical data on the patient, such as pain, disability and clinical course. 

Author Response

The manuscript should be improved. There is a lack of relevant clinical data on the patient, such as pain, disability and clinical course

 -The following sentences were added to the second paragraph of “Case Presentation.”

On physical examination, she had pain in the scapular region when flexing her right shoulder and the intensity was 1-2 as measured by an 11-point pain intensity numeric rating scale. However, the right shoulder was not limited in its range of motion.

Reviewer 2 Report

A very interesting article that reports the case of a scapulothoracic bursitis in a patient with SLE. The manuscript is well written with all up-to-date literature references. All provided images are of adequate quality.

Author Response

A very interesting article that reports the case of a scapulothoracic bursitis in a patient with SLE. The manuscript is well written with all up-to-date literature references. All provided images are of adequate quality.

-Thank you for your consideration.

Reviewer 3 Report

The present manuscript describes an interesting case report about the surgical management of scapulothoracic bursitis in a patient with systemic lupus erythematosus. The aforementioned shoulder disease is quite challenging to manage in daily practice and; in this sense, a detailed description of the diagnosis, treatment, and follow-up can be useful for the readers. Minor revisions are suggested before the acceptance:

- considering the surgical excision of the synovial mass, we suggest the authors revise the title to: "Surgical management of scapulothoracic bursitis in a patient with systemic lupus erythematosus: a case report"

- if possible, we suggest the authors add some histological images of scapulothoracic bursitis. Indeed, considering the diagnosis of systemic lupus erythematosus, we strongly suggest better describing the histological findings. Hyperplasia of the intimal layer? Cellular infiltration of the subintimal layer (leucocytes, lymphocytes, etc)? Delamination of the intimal layer? Subintimal fibrosis? Others? Please for the histological description of the case report refer to Pathol Res Pract. 2022 Dec 12;241:154273. doi: 10.1016/j.prp.2022.154273. Epub ahead of print. PMID: 36563558. 

- considering the peculiar anatomical location of this synovial bursa it's quite challenging to accurately scan it using the ultrasound probe. Often, it's necessary to perform dynamic scanning during active/passive shoulder movements to "uncover" the thoracic wall under the scapula. If the authors have used specific static/dynamic scanning protocols to assess this anatomical area, they can describe them in the manuscript. For a standardized dynamic scanning protocol of the shoulder please refer to Am J Phys Med Rehabil. 2022 Mar 1;101(3):e29-e36. doi: 10.1097/PHM.0000000000001833. PMID: 34923500.  

- did the patient perform a rehabilitation treatment after the surgical excisions of the synovial masses to restore a regular motion of the shoulder? 

Author Response

The present manuscript describes an interesting case report about the surgical management of scapulothoracic bursitis in a patient with systemic lupus erythematosus. The aforementioned shoulder disease is quite challenging to manage in daily practice and; in this sense, a detailed description of the diagnosis, treatment, and follow-up can be useful for the readers. Minor revisions are suggested before the acceptance:

- considering the surgical excision of the synovial mass, we suggest the authors revise the title to: "Surgical management of scapulothoracic bursitis in a patient with systemic lupus erythematosus: a case report"

- Based on your suggestion, we have changed the title to “Surgical management of scapulothoracic bursitis in a patient with systemic lupus erythematosus: a case report.”

- if possible, we suggest the authors add some histological images of scapulothoracic bursitis. Indeed, considering the diagnosis of systemic lupus erythematosus, we strongly suggest better describing the histological findings. Hyperplasia of the intimal layer? Cellular infiltration of the subintimal layer (leucocytes, lymphocytes, etc)? Delamination of the intimal layer? Subintimal fibrosis? Others? Please for the histological description of the case report refer to Pathol Res Pract. 2022 Dec 12;241:154273. doi: 10.1016/j.prp.2022.154273. Epub ahead of print. PMID: 36563558.

- Due to the circumstance of our pathology department, it is not possible to obtain and submit pathology images and findings within 10 days. Please extend the deadline for submitting the revision file.

- considering the peculiar anatomical location of this synovial bursa it's quite challenging to accurately scan it using the ultrasound probe. Often, it's necessary to perform dynamic scanning during active/passive shoulder movements to "uncover" the thoracic wall under the scapula. If the authors have used specific static/dynamic scanning protocols to assess this anatomical area, they can describe them in the manuscript. For a standardized dynamic scanning protocol of the shoulder please refer to Am J Phys Med Rehabil. 2022 Mar 1;101(3):e29-e36. doi: 10.1097/PHM.0000000000001833. PMID: 34923500.  

- The following paragraph was added to the “Discussion.”

Considering the unique anatomical location of the scapulothoracic bursa, it is very difficult to accurately scan it with an ultrasound probe. It is often necessary to perform dynamic ultrasound scanning during active/passive shoulder movements to uncover the thoracic wall under the scapula. Therefore, dynamic ultrasound protocols for the shoulder have been introduced.[11] However, in our case, additional dynamic scans were not performed because MRI is a better diagnostic tool than ultrasound for determining the anatomy of the scapulothoracic region and diagnosing scapulothoracic bursitis.

- did the patient perform a rehabilitation treatment after the surgical excisions of the synovial masses to restore a regular motion of the shoulder? 

- The following sentence was added to the third paragraph of “Case Presentation.”

Despite two surgical excisions, she did not undergo rehabilitation treatment as her shoulder movement remained unrestricted.

Round 2

Reviewer 1 Report

No comments